# Mesenchymal Stem Cell Studies in the Goat Model for Biomedical Research—A Review of the Scientific Literature

**DOI:** 10.3390/biology11091276

**Published:** 2022-08-27

**Authors:** Inês E. Dias, Carlos A. Viegas, João F. Requicha, Maria J. Saavedra, Jorge M. Azevedo, Pedro P. Carvalho, Isabel R. Dias

**Affiliations:** 1CITAB—Centre for the Research and Technology of Agro-Environmental and Biological Sciences, Universidade de Trás-os-Montes e Alto Douro (UTAD), Quinta de Prados, 5000-801 Vila Real, Portugal; 2Inov4Agro—Institute for Innovation, Capacity Building and Sustainability of Agri-Food Production, 5000-801 Vila Real, Portugal; 3Department of Veterinary Sciences, School of Agricultural and Veterinary Sciences (ECAV), UTAD, Quinta de Prados, 5000-801 Vila Real, Portugal; 4CECAV—Centre for Animal Sciences and Veterinary Studies, UTAD, Quinta de Prados, 5000-801 Vila Real, Portugal; 5AL4AnimalS—Associate Laboratory for Animal and Veterinary Sciences, 1300-477 Lisboa, Portugal; 6Department of Animal Science, ECAV, UTAD, Quinta de Prados, 5000-801 Vila Real, Portugal; 7CIVG—Vasco da Gama Research Center, University School Vasco da Gama (EUVG), Av. José R. Sousa Fernandes, Campus Universitário, Lordemão, 3020-210 Coimbra, Portugal; 8Vetherapy—Research and Development in Biotechnology, 3020-210 Coimbra, Portugal

**Keywords:** mesenchymal stem cells, caprine model, MSCs, goat, regenerative medicine, tissue engineering

## Abstract

**Simple Summary:**

This review article aims to compile the works published in the scientific literature, over the last two decades, that use the goat as an animal model in preclinical studies using stem cells, alone or associated with biomaterials, for the treatment of injury or disease in divers organ systems. These preclinical studies are performed prior to human clinical trials for the implementation of new medical or surgical therapies in clinical practice. Thus, it appears that, in the area of tissue engineering and regenerative medicine, the caprine model is particularly used in studies using stem cells in the musculoskeletal system but, although in a more limited way, also in the field of dermatology, ophthalmology, dentistry, pneumology, cardiology, and urology. It appears that the goat represents a particularly useful animal model for studies related to the locomotor system because of its size, and also because they have a more active behavior than sheep, being more similar to the human species in this aspect. Additionally, the goat knee anatomy and the thickness of the cartilage that covers this joint are closer to that of humans than that of other large animal models commonly used in orthopedic research.

**Abstract:**

Mesenchymal stem cells (MSCs) are multipotent cells, defined by their ability to self-renew, while maintaining the capacity to differentiate into different cellular lineages, presumably from their own germinal layer. MSCs therapy is based on its anti-inflammatory, immunomodulatory, and regenerative potential. Firstly, they can differentiate into the target cell type, allowing them to regenerate the damaged area. Secondly, they have a great immunomodulatory capacity through paracrine effects (by secreting several cytokines and growth factors to adjacent cells) and by cell-to-cell contact, leading to vascularization, cellular proliferation in wounded tissues, and reducing inflammation. Currently, MSCs are being widely investigated for numerous tissue engineering and regenerative medicine applications. Appropriate animal models are crucial for the development and evaluation of regenerative medicine-based treatments and eventual treatments for debilitating diseases with the hope of application in upcoming human clinical trials. Here, we summarize the latest research focused on studying the biological and therapeutic potential of MSCs in the goat model, namely in the fields of orthopedics, dermatology, ophthalmology, dentistry, pneumology, cardiology, and urology fields.

## 1. Introduction

Research in stem cell therapy continues to grow, with promising results and increasing expectations in the scientific community. Stem cells can be classified as embryonic stem cells (ESCs), induced pluripotent stem cells (iPSCs), and mesenchymal stem cells (MSCs), which differ in origin, plasticity, differentiation potential, and risk of tumorigenesis [1,2,3,4,5]. MSCs have been the most studied cells, with excellent and safe results in multiple areas. The therapy relies on the anti-inflammatory, immunosuppressive and regenerative properties of these cells [1,6]. Nowadays, different diseases in humans and animals have already been treated with cell-based therapies, mainly to regenerate damaged tissue or reduce inflammation. Nevertheless, there are still some concerns regarding the use of this stem cell therapy and unwanted side effects due to the migration of transplanted cells, as well as insufficient cell survival upon transplantation [7,8]. A solution to this problem might lie in the improvement of animal models and the selection of appropriate methods for grafting and transplantation, facilitating the eventual use of stem cells in clinical practice [8,9,10,11,12]. 

Animal models have been increasingly relevant for human medicine, namely to understand the pathophysiology of diseases and for the development and testing of new therapies [13,14]. The most frequently used animal models are represented by rodents (mice, rats, hamsters, guinea pigs) and lagomorphs (rabbits). Relatively to laboratory animal models, they are generally used and highly contribute to pilot and proof of concept in biomedical research studies. However, they have some important limitations, since they differ substantially from humans in body, organ size, and life expectancy, resulting in huge metabolic, physiological and behavioral differences [13,14]. To overcome this issue, after the first studies were carried out on laboratory animals and to complement them, other larger domestic animal species have been used, such as dogs, pigs, small ruminants (sheep, goats), and horses [14], with promising results. These preclinical translational studies’ results are closer to those expected in human patients. The term “large animal” is used to define animal species that are not included in the laboratory animal models, such as rodents and rabbits [14,15]. As gene sequencing and manipulation tools are now accessible to farm animals (cattle, pigs, sheep, and goats), genetically modified livestock models are available for biomedical research [13].

The use of large animals in these research studies presents great advantages related to their anatomy and dimensions, as they are larger than conventional laboratory animals, being quite similar to humans in most physiological systems [9,16]. Particularly, regarding small ruminants, they are recognized animal models for preclinical and translational studies for human biomedical research. They have the great advantage of being cooperative and compliant, easily available, easy to handle and house, and relatively low cost. They also do not require excessive care with their feeding and sanitary prophylaxis. Their physical stature and weight allow biomedical research to be conducted in more realistic clinical situations. They are livestock species, so their use in experimental research is relatively well accepted by the general society. Sheep have been widely used for biomedical research purposes, resulting in numerous scientific articles and very recent reviews. Comparatively, the goat remains largely unexplored in terms of scientific publications. In biomedical research the goat model has been used mainly for the study of human diseases (chronic rheumatoid arthritis, congenital myotonia, cardiovascular conditions such as atrial fibrillation, and Q fever), as antibody producer (immunological research, immunotherapy, diagnosis), orthopedic, reproduction and cancer research, genome engineering for the production of pharmaceutical proteins, generation of models of human diseases and hosts for the growth of human organs, among other fields. Small ruminants (mainly sheep and goats) have recently been used, namely for regenerative therapy studies with stem cells, in orthopedic and myocardial infarction research [17,18,19]. Nevertheless, large animals still represent less than 0.5% of animals used in stem cell research, mainly due to poor knowledge of the processes responsible for maintaining the pluripotency of stem cells [20]. In view of the above, the aim of this review is to provide a critical overview, discussing the advantages and limitations, and the current state and future perspectives of MSCs therapy studies performed in the goat model, particularly in the orthopedic, dermatology, ophthalmology, dentistry, pneumology, cardiology, and urology fields. A scientific literature search was carried out in the main scientific publication dissemination electronic databases—PubMed, Scopus, and Web of Science. For all the studies referring to the association of the following keywords—goat, stem cells, and animal model in their article title, abstract, or keywords, data were extracted, analyzed, and discussed. The articles published in the English language within the period between January 2000 to June 2022, with the caprine model used in preclinical testing to study stem cell therapies, including the application of MSCs and or hybrid constructs with MSCs seeded onto biomaterials, were reviewed. Only one preclinical study published before the year 2000 was used, as this constitutes the first reference in the goat model of the use of MSCs in cartilage repair [21]. The articles included in this review were all original preclinical and translational studies using MSCs therapy in the goat model in its different organ systems. All other types of articles were excluded, namely, retrospective analyses and literature published prior to 2000.

## 2. The Goat Model

The goat is one of the earliest domesticated animals in the world, being an important part of human culture. Their compact size (compared to cows) makes them attractive from herd management and milking standpoint. Nowadays, there are more than 300 breeds of domestic goats in the world, which provide essential sources of meat, milk, and fiber [16,22]. Cashmere goats are one of the most acclaimed breeds for their annual cashmere fiber production. Recent studies explore the mechanism underlying cashmere growth fiber with cellular models to enrich and optimize follicular cell lines in vitro [23].

Large animal models, namely small ruminants, have the advantages of joint size, bone and cartilaginous thickness most comparable to those of humans, also exhibiting secondary Haversian bone remodeling in the skeletally mature animals, bone tissue macro- and microstructure and composition, bone biochemical properties and bone mineral density more similar to humans [24,25,26,27,28,29,30,31,32,33].

Nonetheless, small ruminants also have remarkable differences compared to humans, namely their rapid growth, with a predominance of plexiform or lamellar bone in areas adjacent to the periosteum and endosteum of long bone cortices during the first years of their lifespan, and the quadruped locomotion [32]. Compared to sheep, the goat is a much higher energy level species, particularly with regard to locomotion, which more closely resembles human musculoskeletal system activity, especially in the study of orthopedic disorders [16,32]. This similarity with humans led to the frequent choice of the goat model for research studies of bone and cartilage repair and hip arthroplasty, although the sheep is also widely used in this type of study. The skeletal maturity is comparable to sheep, reaching around 2–3 years of age [24]. Its anatomical characteristics also make the goat an interesting model for vascular studies. Their long neck with little adipose tissue and large blood vessels allows the exposure and characterization of large jugular veins [16]. The use of small ruminants, namely goats, as models for human and animal research, has some advantages and disadvantages compared to laboratory animals, as described in Table 1. The main advantages are based on anatomical dimensions and physiological similarities, with the exception of the gastrointestinal system, since small ruminants have a polygastric stomach and are herbivores. Furthermore, they are seasonally polyestrous, starting a 10-month period of estrous cycling when daylight hours decreased in autumn [34]. Additionally, at a cellular and molecular level, it should be referred the increasing knowledge of the differences and similarities of MSCs surface marker expression patterns across the human and animal model species, namely the goat [35], and a nearly complete reference genome for goat species, including both sex chromosomes (X and Y) and the autosomes, recently sequenced with high-resolution quality [36]. The longer life expectancy and fewer ethical complications, make it a good preclinical and translational model when compared to laboratory animal models. Some disadvantages lie in the maintenance cost (despite being low, it is still higher than rodents), long gestation time, and the fact that small ruminants are more difficult to handle [8,9,16]. Additionally, the very specific and demanding regulation for the use and maintenance of large animal models may also constitute an added difficulty in carrying out experimental studies in the goat species.

Due to the above-mentioned facts, after the large-scale research phases of therapeutic technologies development, when in vivo testing is a requirement and laboratory animals are used (e.g., mouse, rat, rabbit), the goat model is one of the most used large animal models for preclinical preparations before the clinical trials.

## 3. Overview of Mesenchymal Stem Cells

Mesenchymal stem cells (MSCs) are undifferentiated cells of non-hematopoietic origin, with self-renewal capacity, located in various adult or extra-embryonic tissues. These cells are classified as multipotential, meaning they are capable to differentiate into multiple cell lines, namely cardiomyocytes, chondroblasts, endothelial cells, hepatocytes, myocytes, neuronal cells, osteoblasts, and tenocytes, among others [1,2].

MSCs can be obtained from embryonic cells in the first stages of embryo development before its implantation in the uterus, and in adults, they can be isolated from various tissues, such as umbilical cord, placenta, amnion, bone marrow (BMSCs), adipose tissue (ASCs), dental pulp and periosteum [37,38,39]. The diversity of sources of stem cells or MSCs and the wide range of potential applications of these cells lead to a challenge in selecting a suitable cell type for cell therapy [8]. In veterinary medicine, MSCs are collected mainly from bone marrow (BM) or adipose tissue (AT), since these tissues are easier to obtain [2,4,40].

Depending on how MSCs are obtained, therapy with these cells can be autologous, if they are obtained from the same animal, allogeneic if the donor is another individual from the same species, or xenogeneic if the donor is from a different species. Regarding the route of administration, they can be applied locally, systemically, generally intravenously, or both, depending on the disease [37].

These cells have the ability to differentiate into the target cell type allowing the regeneration of the injured area and also have great immunomodulatory potential. This immunomodulation is due to paracrine effects (by secreting different cytokines and growth and differentiation factors (GDFs) to adjacent cells) and by cell-to-cell contact, leading to vascularization, cell proliferation in injured tissues, and reducing inflammation [1,6,37,41]. Different diseases in animals have been treated using cell-based therapy, mainly to regenerate damaged tissue or reduce inflammation. These cells are considered “immune privileged” as they do not express histocompatibility complex class II (MHC-II) and costimulatory molecules (such as CD40, CD80, and CD86), allowing allogeneic therapy, as they escape the recognition and action of T cells and NK receptors [1,2,6,42].

MSCs can be applied in a wide range of clinical specialties, such as traumatology and orthopedic, ophthalmology, neurology, internal medicine, dermatology, and immunopathology, among others. Figure 1 shows the medical specialties studied for MSCs therapy in the goat model. The applications of these therapies in the veterinary medicine field include not only their clinical use in domestic animals, but also the translation of the results, as a preclinical model, for human medicine [14]. These cells show great resistance to cryopreservation, allowing the creation of cell banks for later use and the selection of the best donors by previously evaluating their MSCs in vitro [2].

## 4. Overview of Biomaterials for MSCs Regenerative Medicine and Tissue Engineering

As an important part of tissue engineering and regenerative medicine, materials (natural or synthetic) should be considered to improve the therapeutic efficacy and regenerative potential of MSCs-based therapy and tissue engineering hybrid constructs. Materials can induce or inhibit MSCs adhesion, proliferation, and differentiation, which plays a critical role in MSCs’ therapeutic potential. Functional biomaterials should, therefore, be used to promote proliferation, and specific lineage differentiation and reinforce in vivo survival and engraftment of transplanted MSCs [43,44]. 

The commonly used scaffolds for tissue engineering include hydrogels, electrospun scaffolds, and nano/microspheres [44]. Scaffolds can be derived from natural polymers (e.g., collagen) containing adhesion peptides essential for direct cell adhesion, or derive from synthetic polymers. When using synthetic polymers (e.g., poly(lactic-co-glycolic acid), polycaprolactone), the bioactive molecules that are crucial for cell adhesion, proliferation, and differentiation, are usually obtained by absorption, dipping the scaffolds in serum or cell culture medium [44,45].

Hydrogels are one type of scaffold that have been used in a wide variety of applications, including vascularization, tendon repair, kidney repair, and neural regeneration [44]. Hydrogels based on natural polymers are an option to induce MSCs for chondrogenic differentiation in human and animal models, and also allow the capture of chondroinductive growth factors and cytokines, such as TGF-β1, -β2, and -β3 [43]. Tissue engineering has evolved greatly over the last few years, and scaffold production can be tailored to better suit and mimic the in vivo conditions of where scaffolds are placed. From different coatings that facilitate cell adhesion, to functionalized molecules that are going to guide cell behavior and be released at specific time points, there is ever-growing evidence to support the replacement of living tissues with different biomaterials [45].

## 5. Application of MSCs in the Goat Model

### 5.1. Orthopedics

MSCs are an alternative source of cells for cartilage, muscle, tendon, and bone regeneration, as they are easily accessible and can be harvested from different tissues, having a great capacity to proliferate and differentiate into other sorts of cells in the body, such as osteoblasts and chondroblasts [37,46,47]. Caprine models have been used in preclinical and translational research studies to assess new approaches, resorting to MSCs for cartilage tissue engineering aiming for regenerative joint resurfacing, osteoarthritis treatments, bone fractures and defects, menisci and anterior cruciate ligament injury repair studies. Additionally, several studies have been performed on the goat at the spine level, namely for research on new treatments for intervertebral disc (IVD) disease.

#### 5.1.1. Cartilage Repair

Articular cartilage covers the end of the bone, helping the bones easily glide over each other. Due to its avascular nature and its low cellular density, this tissue has inadequate self-healing capacity. Therefore, in the case of injury-causing cartilage defects, these lesions rarely resolve spontaneously, forming a fibrous tissue with functional properties distinct and inferior to native hyaline cartilage, leading to joint deterioration [1,41,48,49,50]. 

Cartilage lesions are usually treated by chondroplasty and palliative debridement techniques, drilling and microfracture perforation (MF), or restoration techniques using autologous chondrocyte implantation, mosaic arthroplasty and osteochondral allograft transplantation [51,52,53]. Of these several techniques applied to repair cartilage in humans, none of them present guaranteed results. Recent advances in osteochondral tissue engineering and regenerative medicine have been used to promote healing in areas of articular cartilage damage that generally do not respond to more conventional treatments, enhancing the healing of injured tissue so that it returns to its original or near-original condition [37,54]. A variety of studies with transplanted cells—autologous, allogeneic, and xenogeneic MSCs therapy, or novel replacement devices—have shown great results in cartilaginous tissue healing and regeneration [55]. MSCs are able to differentiate into chondrocytes when cultured alone or in combination with GDFs, making them interesting for cartilage regeneration, meniscal repair, osteoarthritis therapy, and also ligaments and tendon repair.

Human cartilage lesions have nearly a mean total volume of 552.25 mm^3^, with at least 10 mm in diameter, and typically just involve the chondral tissue in 95% of clinical presentations [24,25,56,57,58,59]. On the contrary, the mean total volume of the experimentally induced cartilage defects in animal models generally presents a reduced volume compared to human clinical presentations, in the goat at 251.65 mm^3^ [24]. Additionally, frequently these induced defects in animal models involve the subchondral bone, which could cause the subsequent advancement of the subchondral bone plate during spontaneous healing of osteochondral defects and following articular cartilage treatment for chondral lesions in various research studies with animal models [60]. This fact could be partly justified by the marked difference in cartilage thickness between human and animal model species, where humans have the thickest articular cartilage at the stifle joint level—2.35 mm [61]. The thickness of the medial condyle cartilage in goats has been reported by Brehm et al. (2006) ranging between 0.8–2 mm and by Frisbie et al. (2006) with 1.1 mm resulting in extensive variations of the chondral and subchondral bone volume involved in different studies [62,63]. In this way, choosing an appropriate animal model for chondral or osteochondral tissue engineering studies should be based on published scientific literature, guideline documents from the American Society for Testing and Materials, the International Cartilage Repair Society, and the US Food and Drug Administration (FDA) [64,65].

The goat stifles joint anatomy and the cartilage articular thickness is similar to humans, thus allowing arthroscopic examination and the creation of chondral defects, without affecting the osteochondral bone plate, but has a limited capacity to heal without iatrogenic intervention, like in humans [24,25,59,63]. Table 2 presents the main MSCs studies performed in the caprine model for cartilage repair.

Studies with the goat model for cartilage repair have demonstrated success in therapy with fetal progenitor cells and Wharton’s jelly derived mesenchymal stem cells (hWJMSCs) [69] and BMSCs alone or associated with scaffolds or GDFs [69,71,72]. Zhang et al. (2018) studied the potential of the hWJMSCs in a caprine model with a full-thickness femoral condyle articular cartilage defect, compared with the MF technique. The results showed that the hWJMSCs improved higher quality of hyaline cartilage regeneration, maintaining the structure and functional integrity of the subchondral bone, compared to MF [69]. Animals showed significant improvement, but the clinical results came better when associated with collagen, primary cartilage cells, chondrons, or hyaluronic acid [71]. In another study, gene therapy was performed by the application of human TGF-β1 gene-transduced autologous BMSCs, in the sodium alginate and calcium chloride (CaCl_2_) to create calcium alginate gels, associated with mosaic arthroplasty [68].

Li et al. (2021) demonstrated that the association of BMSCs and collagen type I have better results in goat knee cartilage defect models, than either one alone [71]. 

#### 5.1.2. Osteoarthritis

As a consequence of cartilage degeneration, osteoarthritis (OA) may arise. OA is a degenerative and inflammatory disease that affects all the joint tissues (synovial membrane, bone, cartilage, meniscus, ligament, tendon), resulting in loss of articular cartilage, the release of inflammatory and regulatory cytokines, leading to pain and lameness [37,49,73]. The cartilage inside a joint begins to wear down and the underlying bone starts to change [37,49]. Due to the lack of any etiologic treatment that could stop or delay the changes in the joint tissues, the current treatment for patients with OA is mainly based on the use of analgesics and anti-inflammatory drugs. More recently, thanks to a better understanding of OA pathophysiology, new therapeutic approaches have emerged [73]. Intraarticular injection of autologous platelet-rich plasma (PRP) [74,75,76] and MSCs have recently gained special attention as promising tools for OA treatment [77,78] and can be used separately or as an association therapy. PRP acts as a scaffold, and through the release of GDFs (such as transforming growth factor beta (TGF-β), fibroblast growth factor (FGF), and insulin-like growth factor (IGF-1)), induces stimulation of chondrogenesis, increases hyaluronic acid production, develops angiogenesis and leads to differentiation of the existing cells in the treated zone [37,79,80]. MSCs are also a promising therapy as they differentiate into chondrocytes once they are cultured alone or in combination with GDFs [37,41,81,82].

The goat model has also been used to investigate stem cell therapy after induced OA by meniscectomy in association or not with anterior cruciate ligament (ACL) resection. Specifically, in the study of Murphy et al. (2003), a single dose of ten million BMSCs suspended in a diluted solution of sodium hyaluronan was directly injected into the injured stifle joint 6 weeks after OA induction and after 6 and 20 weeks of the operative period [83]. In another recent study, a single intra-articular dose of 7 × 10^6^ human ASCs was injected 9 weeks after the meniscectomy associated with daily injections of cyclosporin A (10 mg/kg for 7 days followed by 5 mg/kg for another 7 days) was applied and evaluated 8 weeks after injection of human ASCs [84]. A study conducted by Wang et al. (2018) showed that BMSCs therapy in a goat OA model had a greater benefit in terms of cartilage protection when compared with PRP therapy [85].

#### 5.1.3. Meniscal Repair

Meniscus repair also remains a challenge in orthopedics. It is formed by a semilunar fibrocartilage structure that is essential to maintain normal stifle joint function [86]. Similar to the articular cartilage, the meniscus is poorly vascularized in the inner area [87,88] and there is no ideal reconstructive approach for damaged menisci [88]. GDFs, scaffolds, and hybrid constructs, with a resource for static/dynamic cell cultures, can be combined for meniscal tissue repair [88].

Caprine models with large radial tears of the meniscus have also been used to study meniscal repair. Rothrauff et al. (2019) verified by magnetic resonance image and by macroscopic and histological scoring, that the stromal vascular fraction (SVF), which contains, among other cell types, MSCs, seeded in a hydrogel contributes to the successful repair of meniscal tears [89]. Gene therapy also provides one promising minimal invasive alternative strategy for meniscus. Zhang et al. (2009) used BMSCs with the transfection of the human IGF-1 (hIGF-1) gene, one of the most important GDFs in cartilage homeostasis and development. The author verified promising results to repair defects, especially when combined with calcium alginate gel [88].

#### 5.1.4. Anterior Cruciate Ligament Injury Repair

ACL is the main structure that maintains the stability of the knee [90]. When ruptured, usually as a sports injury, provokes tibia internal rotation and instability of the stifle joint, resulting in lameness, pain, inflammation, and, in the long run, it leads to OA [37,91,92]. In veterinary medicine, the therapeutic approach is not consensual among authors. Some advocate surgical therapy as the first choice in all patients, with alternate methods of ACL repair, through extracapsular surgical techniques or surgical procedures which promote changes in stifle joint anatomy and biomechanics, promoting the dynamic stability of the knee [93]. However, no specific surgical technique has yet been defined as the ideal standard in veterinary medicine.

Grafts, including autografts, allografts, and synthetic grafts are routinely used to reconstruct ACL rupture [90]. Zhao et al. (2015) studied the feasibility of biological xenogeneic ligament graft in a goat model combined with BMSCs for the reconstruction of ACL. The results showed that there was no immune rejection with the xenogeneic graft [94]. Research showed that xenogeneic ligament combined with BMSCs can accelerate microcirculation and lead to ligament growth, significantly improving ligament revascularization, without influencing the biological characteristics of the ligament [94].

#### 5.1.5. Bone Fracture and Defect Repair

The repair of bone fractures and segmental bone defects secondary to trauma, post-tumor resection, or post-debridement infection remains a major clinical problem, usually implying a large economic burden [37,41,95,96,97]. The use of animal models, such as goats, is important to develop bone tissue engineering approaches, mimicking real clinical conditions in humans [97,98]. Small ruminants have similar body weights and compatible long bone sizes for studying human implants and prostheses [99].

Currently, therapeutic methods for bone critical size defects (CSDs) include autografts, allografts, and synthetic bone grafts [100]. A promising alternative to autologous bone grafts is the combination of BMSCs with porous osteoconductive scaffolds [101,102]. For this purpose, BMSCs cells are usually isolated, expanded in vitro, and seeded in a 3D porous scaffold. This construct is then implanted into the bone defect to achieve in situ regeneration [100].

To treat bone CSDs, an effective bone substitute is essential [103,104]. Bone substitutes frequently used in clinical practice lack efficacy and have a low osteogenic capacity [70,103,104]. An ideal scaffold should be biocompatible, biodegradable, and promote the passage of nutrients and cellular waste products [105,106]. Furthermore, vascularization is essential for optimal oxygen and nutrient supply of seeded cells [101,107]. Noteworthy, the addition of endothelial progenitor cells (EPCs) to MSCs culture improves vascularization and increases bone formation [107].

To achieve a better efficacy for the treatment of bone CSDs, several bone substitutes and scaffolds have been studied in the goat model for femoral, tibial, and other bone defects (Table 3). 

Small ruminants have also been used as an animal model to study the pathophysiology of osteopenia or post-menopausal osteoporosis [116]. Goats provide a promising model for studying osteoporosis caused by a lack of estrogen [117]. A study carried out by Cao et al. (2012) demonstrated that estrogen deficiency is an obstacle to cell therapy for bone regeneration. This study proved that β-TCP with autologous BMSCs as a bone substitute has successfully repaired critical size bone defects in the femur of osteoporotic goats [117].

Recent studies in goats also combined tissue-engineered bone and gene therapy to treat critical size bone defects. They used BMSCs transduced with human morphogenetic protein-2 (hBMP-2) to provide osteoprogenitor cells, osteoinductive factors, and osteoconductive carriers, improving healing capacity. The results were better in the groups treated with BMSCs combined with hBMP-2 than those treated with BMSCs alone [118].

#### 5.1.6. Vertebral Column

Intervertebral disc (IVD) disease is a chronic progressive and painful disease that affects millions of people worldwide. This disease is one of the most common causes of low back pain [119,120]. IVD disease typically begins with tears in the outer ring of the IVD (annulus fibrosus), which can lead to a reduction in the water content of the soft gel center of the disc (nucleus pulposus). The wear and tear of IVDs may result from normal aging or may be due to long-standing trauma [119,120].

Although no animal model could completely reproduce the clinical conditions in humans, large animal models are preferable since in small rodents or rabbits it is more difficult to administer cells into the disc due to the very small size of the IVDs [121]. Goats represent a suitable model, as their discs are similar in shape and size to humans [121,122].

Currently, treatment options for this disease, such as physical rehabilitation, pain management, and surgical intervention for disc decompression, provide only temporary pain relief [122]. Surgical techniques include spinal fusion to decrease pain and neurological deficit in selected patients. To decrease excessive spinal motion and restore proper alignment and intervertebral height, a spinal fusion should be considered. To achieve spinal fusion, several techniques have been proposed, including the use of an interbody spacer (cage) [123], which can be filled with autologous iliac crest bone graft (ABG) or substitute scaffolds. It is likely that the poly(L-lactide-co-caprolactone) (PLCL) scaffold binds to ASCs, which rapidly proliferate and lead to their differentiation into osteocytes [124]. However, Kroeze et al. (2014) demonstrated that while the addition of stem cells to the PLCL scaffolds did not result in adverse effects, it also did not increase the rate and number of interbody fusions. Future studies are needed to optimize the spinal fusion model with PLCL scaffolds and MSCs [123].

Other new minimally invasive disc therapeutic approaches are being studied, such as the combination of hydrogels, MSCs, and PRP [125,126]. Unlike MSCs, studies with goat models of IVD disease with stromal vascular fraction (SVF) therapy have been contradictory, with some inflammatory adverse effects [127]. Table 4 presents the main MSCs studies performed in the caprine model for vertebral column repair.

### 5.2. Dermatology

Chronic cutaneous wounds and ulcers represent a therapeutic challenge, due to the difficulty of clinical management, high recumbence rate, and scar formation, both in human and veterinary medicine. Its incidence has increased due to population aging, diabetes, obesity, and concomitant diseases [2].

MSCs present an important role in all phases of tissue repair: inflammation, proliferation, and remodeling [2,3]. These cells also promote angiogenesis and show evidence of antimicrobial properties. A failure in the angiogenesis process during wound healing can induce the development of chronic wounds [2,3]. MSCs have already been shown to promote wound healing through a paracrine molecular cascade in goat models [128,129]. Furthermore, MSCs may have direct involvement in various stages of the wound healing process that need to be further explored [129].

Although goats have not generally been used as research models for cutaneous wound healing, they are a good choice as they have a mild temperament and a good anatomical skin surface available for creating lesion models of different shapes and sizes [16].

Studies with the goat model for wound healing have used different sources of MSCs with excellent results. Pratheesh et al. (2017) showed evidence of better wound healing with MSCs from the amniotic fluid origin than caprine BMSCs, by revealing greater epithelialization, neovascularization, and collagen development in the histomorphometric analysis [130]. Azari et al. (2011) also showed the re-epithelization capacity of transplanted Wharton’s jelly MSCs from caprine umbilical cords, revealing complete re-epithelization of cutaneous wounds in 7 days [131].

In addition, MSCs showed great capacity for wound regeneration and reduced healing time and plasticity, as they are capable of converting into cells of different tissues. A study carried out by Yang et al. (2007) with goats confirmed that epidermal adult stem cells can differentiate into different functional cells in vivo or in vitro, demonstrating the plasticity of stem cells [132].

### 5.3. Ophthalmology

In recent years, ophthalmologists have placed a great focus on stem cells to treat various traumatic and degenerative disorders due to their unique biologic properties [133]. The cornea is a protective barrier and is formed by three layers with different germinal origins: the epithelium (originated from superficial ectoderm) and the stroma and endothelium (originated from neural crest cells) [134]. Experimental studies have proved that there is a variety of stem cells present in each of these layers [134,135]. For example, limbal stem cells (residing in the limbus) maintain epithelial homeostasis and regenerate the cornea, with epithelial cell deficiency being the leading cause of blindness worldwide [134].

Stem cells have high potential in the treatment of eye diseases characterized by permanent cell loss, such as glaucoma, age-related macular degeneration, photoreceptor cell degeneration, hereditary retinopathy, and mechanical and ischemic retinal injuries [134,136,137]. The eyes of small ruminants are anatomically different from human eyes [16,138], however the resemblance in structure, size with, some properties and parameters to the human eye made possible to use these models successfully. Goats have been used mostly to study corneal epithelium reconstruction and transplant. 

Studies have shown that epidermal adult stem cells (EpiASCs) from goat ear skin can be used to successfully repair damaged cornea with total limbal stem cells (LSCs) deficiency [139,140]. Moreover, these results demonstrated that EpiASCs can be induced to differentiate into corneal epithelial cell types in vivo in a corneal microenvironment, and had the skill to trigger corneal genetic programs [139].

In a study carried out by Mi et al. (2008), cryopreserved limbal corneal stem cells were applied in goats with damaged cornea with excellent results. The therapeutic effect of transplantation may be associated with the inhibition of inflammation-related angiogenesis after transplantation of cryopreserved LSCs [141].

### 5.4. Dentistry

There are only a few published studies with goats for stem cell therapy in dentistry, probably due to the anatomical differences between human and goat dentition. Because it is a ruminant species, it only has incisors in the mandible, being absent in the upper jaw. It has no canines and only premolars and molars, the latter fulfills the function of rumination of plant foods. Stem cell research in dentistry aims at the regeneration of damaged tissues such as periodontal tissues, dentin, pulp, and resorbed roots and the repair of endodontic iatrogenic perforations [142]. It is mainly used for periodontal regeneration and in association with biomaterials to optimize tooth regeneration in the goat model [143,144]. Undifferentiated MSCs are able to differentiate, providing the three critical tissues essential for periodontal tissue regeneration: cementum, bone, and periodontal ligaments, making stem cells a new approach for periodontal tissue regeneration [145]. The association of MSCs and fibrin glue using particulate mineralized bone has also shown promising results for vertical bone augmentation in animal models [146].

There is a growing need to use dental implants and improve their function to enhance normal dental physiology and proprioception [142]. An osseointegrated implant can closely resemble a natural tooth. Nevertheless, the absence of periodontal ligament and connective tissue results in important differences in implant adaptation to occlusal forces [145]. Several studies for tooth regeneration with scaffolds based on biomaterials and stem cells have shown very positive effects on regeneration [143,144,145,147]. Dense collagen gel scaffolds seeded on MSCs and nanostructured titanium surfaces have increased interest in bone regeneration due to the good osteointegration effect [148,149].

The placement of implants can be problematic, as, after a tooth loss, anatomical pneumatization of the maxillary sinuses can occur, as well as atrophy of the alveolar ridge, limiting the bone volume available to the implant placement. Maxillary sinus floor elevation is one of the preferred surgery options to solve this problem, where bone graft material is placed in the maxillary sinus to provide adequate support to the implants [143]. Zou et al. (2012) associated this grafting material with BMSCs and calcium phosphate cementum, promoting earlier bone formation and mineralization, and maintaining the height and volume of the augmented maxillary sinus in a goat model [143].

Bangun et al. (2021) proved that MSCs can improve earlier bone repopulation and complete faster bone regeneration in tissue-engineered bone grafts, supported by the paracrine activity of the resident stem cells [144].

It is also known that TGF-β1 plays an important role during tooth formation, this GDF can directly induce the differentiation of odontoblast-like cells, and positively regulate the secretion of matrix components in the dentin-pulp complex, being a potential therapy to induce tissue formation after dental pulp capping treatments [150]. 

### 5.5. Pneumology

Stem cells have been a promising therapy for asthma non-responsive to conventional therapy [1,42], as a potential treatment for destructive lung diseases including chronic obstructive pulmonary disease [151], and as a treatment for bronchopleural fistula (BPF), due to its plasticity and ability to differentiate into different cells [152].

Goats have been successfully used in a bronchopleural fistula model. In cases of lung cancer with limited disease, the most effective method of controlling the primary tumor is surgical resection, as it offers the best chance of cure. Pulmonary resection can lead to the development of a pathological connection between the airway (bronchus) and the pleural space, known as a post-resection BPF [152]. This research proved that bronchoscopic-guided transplantation of BMSCs successfully closes bronchopleural fistula by extraluminal fibroblast proliferation and collagenous matrix development [152].

### 5.6. Cardiology

The goat seems to be the ideal model for cardiovascular diseases, due to its anatomical dimensions and physiological similarities to the human heart. Additionally, parameters such as heart rate, coronary architecture, and capillary density are more similar to those of man [9,14].

One of the main causes of morbidity and mortality in cardiovascular diseases is myocardial infarction, characterized by the ischemic lesions of cardiac muscle tissue due to an occlusion of one of the coronary arteries or one of its branches by a thrombus [17]. This disease is increasing and gaining more importance due to the general aging of the population and the change in lifestyle [17,18].

The goat has been studied for the treatment of myocardial infarction through in vitro and in vivo studies. So far, several different sources of adult stem cells have been identified as a healing approach for infarcted myocardium. The most promising stem cells that have so far shown the best results in these studies are BMSCs. A study carried out by Liao et al. (2006) uses BMSCs enriched by small intestinal submucosal (SIS) films to treat myocardial infarction in goat models, the MSCs-SIS film was implanted and sutured in the infarct area. The obtained results revealed that this therapy can prevent ventricular chamber dilatation and can improve myocardium contractility, cardiac function, and collateral perfusion [18].

Another promising source of stem cells is glandular stem cells that can be easily extracted from exocrine glands, such as the salivary glands or the pancreas [17,153,154]. Maass et al. (2009) showed that glandular stem cells obtained from the submandibular gland can spontaneously differentiate into cardiac-like mesodermal cells in vitro. These results suggest that implanting these cells directly into infarcted myocardium can improve heart regeneration [17].

### 5.7. Urology

Cell-based therapy is emerging as a great alternative in the treatment of stress urinary incontinence. Goats are also an ideal model to study this subject, as the female caprine urethra has similar parameters to those reported in humans, by measuring the urethral pressure profile, making them a suitable experimental animal for testing intraurethral cell transplantation [155].

Burdzinska et al. (2017) demonstrated that caprine muscle-derived cells (MDCs) and MSCs can be expanded in vitro and applied for intraurethral injections [155]. In 2018, Burdzinska proved that both MSCs and MDCs collaborated in the formation of striated muscle when they were transplanted directly into the external urethral sphincter. This study suggests that MDC-MSC co-transplantation improves urethral closure better than if transplantation of each cell population is performed alone [156].

## 6. Conclusions

MSC-based cell therapy has the potential to treat diseases and injuries with excellent and safe results in several areas. They have the ability to replace damaged cells and modulate the immune system in vivo. These characteristics combined with their ease of isolation, expansion, and manipulation in vitro, make MSCs attractive candidates for numerous therapeutic conditions [14].

The goat model has been studied mainly in the orthopedic field, probably due to its similarity with human stifle joint and also because of its body weight and long bone sizes proportionally compatible with humans. Furthermore, the goat model appears to be extremely useful for the study of cardiac and urologic diseases, such as myocardial infarction and stress urinary incontinence due to the similarity in the anatomy of the goat heart and urethra, respectively, to those of humans. Given the similarity between the goat and man regarding the anatomy of the bones, heart, and urethra, the most promising areas to be studied with the goat model are orthopedics, including cartilage, bone repair and also IVD disease, cardiology, and urology.

Almost all studies in the goat model revealed good to excellent results with MSCs therapy alone or in combination with other therapies, such as PRP, GDFs, biodegradable or nonbiodegradable scaffolds, namely in the regenerative medicine and tissue engineering studies of the musculoskeletal system. One of the preferable sources of MSCs is bone marrow, especially in the orthopedic field, as BMSCs seem to be a better source of cells for bone regeneration [157].

In conclusion, the goat is an appropriate model for studying a wide range of diseases. Further research in large animal models will be needed to ensure safety and efficacy, as well as establish appropriate stem cell therapy protocols, including doses and routes of administration. It is expected that in the future a considerable number of studies using large animal models will be performed to improve regenerative medicine in the veterinary field and also as models for reproducing the disease in preclinical experimental studies, establishing the goat as an important translational model for human medicine application.

## Figures and Tables

**Figure 1 biology-11-01276-f001:**
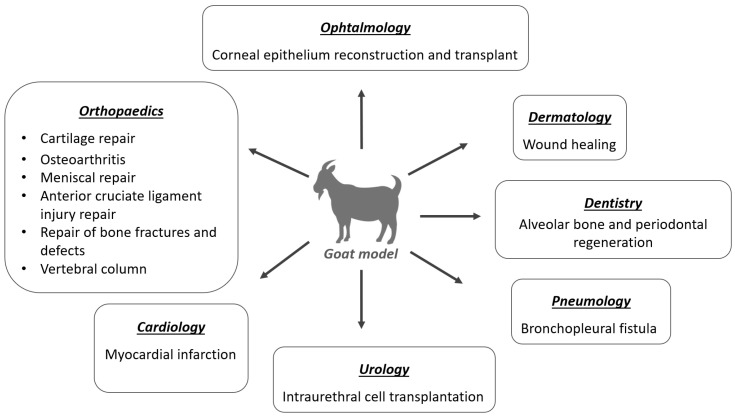
Clinical applications of MSCs studied in the goat model, according to medical specialties.

**Table 1 biology-11-01276-t001:** Advantages and disadvantages of the goat model [8].

Advantages	Disadvantages
Similarity of anatomy and dimensions to humans (most similar to human heart and stifle joint) imilarity of physiology and/or clinical procedures to humans Docile animals Longer life expectancy (7–14 years depending on breed) Fewer ethical issues than with companion animals (dogs and cats) and horses	Higher maintenance costs than rodents Harder to handle, requires specialized housing facilities infrastructure and trained personnel Longer gestation time Differences in the gastrointestinal system, since the goat is a polygastric animal species (ruminant)

**Table 2 biology-11-01276-t002:** MSCs studies in the caprine model for cartilage repair.

Population Sample	Type of Defect and Localization	Material Tested and Follow-Up Period	Reference
Goats	Round 4.0 mm Ø hole; femoral condyle	Auto- and allogeneic BMSCs implantation	Butnariu-Ephrat et al. (1996) [21]
Female Nubian cross goats > 2 years of age	Cylindrical full-thickness osteochondral 4.5 mm Ø × 4.0 mm deep; proximal 1/3 of the medial facet trochlear groove, bordering the axial groove + anterior 1/3 of the central medial femoral condyle	BMSCs + gelatin constructs Timepoints: 1, 2, 7, and 14 days	Quintavalla et al. (2002) [66]
12 goats (4 animals per group)	Osteochondral 6 mm Ø × 12 mm in depth; medial femoral condyle weight-bearing areas	TE osteochondral graft of autologous BMSCs-*β*-TCP scaffold with/without mechanical stimulation of stir and an untreated defect Timepoints: 12 and 24 weeks	Pei et al. (2014) [67]
18 male goats, ±22.5 kg	Cylindrical 9 mm Ø × 3 mm in depth defects; weight-bearing area of the medial femoral condyle	Human TGF-β1 gene-transduced autologous BMSCs in sodium alginate (density of 5 × 10^7^ cells/mL) and 10^2^ mmol/L CaCl_2_ to form calcium alginate gels + mosaic arthroplasty (2 to 4 cylindrical osteochondral 5.5 mm Ø × 3 mm Ø grafts from the medial femoral condyle periphery) Timepoint: 24 weeks	Sun et al. (2016) [68]
6 white male skeletally mature goats (30 ± 5 kg) divided into two groups	Full-thickness articular cartilage defects; femoral condyle (including lateral and medial femoral condyles) with 6.5 mm Ø	hWJMSCs (density 1 × 10^6^ cells) at the third stage seeded onto an ACECM-oriented scaffold (1 mm thickness × 6.5 mm Ø) compared with microfracture technique Timepoint: 9 months	Zhang et al. (2018) [69]
24 goats, 10 months	Osteochondral defect (10 mm Ø × 12 mm depth); femoral heads	Treatment with a biomanufacturing platform constructed with porous tantalum and collagen membrane with/without BMSCs Timepoint: 16 weeks	Wei et al. (2019) [70]

β-TCP—beta-tricalcium phosphate, hWJMSCs—Wharton’s jelly derived mesenchymal stem cells, BMSCs—bone marrow derived-mesenchymal stem cells, TGFβ—transforming growth factor beta.

**Table 3 biology-11-01276-t003:** MSCs studies in the caprine model for bone fracture and defect repair.

Population Sample	Type of Defect and Localization	Material Tested and Follow-Up Period	Reference
36 goats, 14.5–15.5 kg	20 mm long defect of left tibial bone and periosteum	Four groups treated with: - Coral hydroxyapatite, - Coral hydroxyapatite + BMSCs, - Fascia flaps, - Untreated group Timepoints: 2, 4, 8, and 12 weeks	Chen et al. (2006) [108]
6 adult goats, 20–30 kg	Femur hollow cylinder of 2 cm length, 2 cm outer Ø and 7 mm inner hole Ø	Two groups treated with: - bioactive triphasic ceramic-coated HASi + BMSCs, - HASi without cells	Nair et al. (2008) [109]
8 adult female goats, 46.3–22.9 months, 54.1–75 kg	4 non-critical-sized defects 6 mm Ø on the medial diaphyseal tibia	Four groups treated with a polymer of methacrylate-endcapped poly(D,L-lactide-co-e-caprolactone): - Pure polymer with a triacetin solution, - Polymer with a triacetin solution + autologous MSCs, - Polymer with triacetin solution + α-TCP, - Untreated group Timepoints: 2, 4, 8, and 12 weeks	Vertenten et al. (2009) [110]
8 adult goats, 7–8 months, 17–20 kg	Cranial bone defects of 20 mm Ø	Three groups treated with: - BMSCs/PLGA + PRP, - BMSCs/PLGA + PPP, - BMSCs/PLGA + DMEM Timepoints: 3 days and 8 weeks	Lei et al. (2009) [100]
6 adult goats 20–30 kg	2 cm femur bone segment was excised from the mid-diaphyseal region	Three groups treated with: - HASi without cells, - HASi + BMSCs, - HASi + BMSCs + PRP Timepoints: 6 weeks, 2 months	Nair et al. (2009) [111]
8 female goats, 47.3 ± 17.5 months, 66 ± 12 kg	4 unicortical holes (6 mm Ø) on the medial diaphyseal cortex	Polymerizable pluronic f127 hydrogel derivate combined with autologous MSCs with different types of carriers Timepoints: 2, 4, 6, and 8 weeks	Lippens et al. (2010) [112]
32 adult male goats, ±3 years, ±50 kg	25 mm defect on the tibia	Four treated groups: - nHACP/CF (nano-hydroxyapatite/ collagen/poly(L-lactic acid/chitin fibres) + goat BMSCs, - Autograft bone, - nHACP/CF, - Untreated group Timepoints: 4 and 8 weeks	Liu et al. (2010) [113]
6 female goats, 1 year, ±25 kg	30 mm defect on the tibia	Two groups treated with: - β-TCP combined with autologous BMSCs cultured by dynamic perfusion, - β-TCP combined with autologous BMSCs cultured by static perfusion Timepoints: 1, 4, 12, and 24 weeks	Wang et al. (2010) [114]
4 adult female goats, 30–45 kg	8 drill holes (6 mm Ø × 3.0 mm depth) on the lateral diaphysis of both posterior femurs	Cell-scaffold of goat BM stromal cells and SPCL (a blend of starch with polycaprolactone) Timepoints: 2, 4, and 6 weeks	Rodrigues et al. (2011) [105]
6 female goats, 2 years, ±25 kg	42 mm defect in the diaphyseal region of the tibia	Polycaprolactone scaffold seeded with goat BMSCs cultured in a perfusion bioreactor Timepoints: 6 and 12 weeks	Gardel et al. (2014) [115]

BM—bone marrow, SPCL—blend of starch with polycaprolactone, BMSCs—bone marrow derived-mesenchymal stem cells, PLGA—poly(lactic-*co*-glycolic acid), PRP—platelet-rich plasma, PPP—platelet-poor plasma, DMEM—Dulbecco’s modified Eagle’s medium, HASi—bioactive triphasic ceramic-coated hydroxyapatite scaffold, α-TCP—alpha-tricalcium phosphate, nHACP/CF—nano-hydroxyapatite/collagen/poly(L-lactic acid)/ chitin fibers, β-TCP—beta-tricalcium phosphate.

**Table 4 biology-11-01276-t004:** MSCs studies in the caprine model for vertebral column repair.

Population Sample	Type of Defect and Localization	Material Tested and Follow-Up Period	Reference
7 mature female goats	8 mm hole was created through the IVD and the adjacent endplates of L1-2 and L3-4 v ertebrae	Treatment with a spinal cage (10 mm × 10 mm × 18 mm) filled with autologous bone from iliac crest or filled PLCL scaffold seeded or not with SVF Timepoints: 1, 3, and 6 months	Vergroesen et al. (2011) [124]
24 male goats, 4 years	L1-2 and L3-4 vertebrae were injured by an nº 15 blade scalpel with a 15 mm depth	Influence of BMSCs therapy suspended in hydrogel compared with control groups without BMSCs. Stem cells increase proteoglycan levels Timepoints: 3 and 6 months	Zhang et al. (2011) [121]
36 mature female goats, 54–103 kg	8 mm hole was created through the IVD and the adjacent endplates of L1-2 and L3-4 v ertebrae	Treatment with a spinal cage (10 mm × 10 mm × 18 mm) filled with: - autologous bone graft, - PLCL alone, - PLCL seeded with SVF, - PLCL seeded with ASCs Timepoints: 3 and 6 months	Kroeze et al. (2014) [123]
12 female adult goats	Injection of 0.25 U/mL chondroitinase ABC in five lumbar discs through a left retroperitoneal approach	SVF to establish intervertebral disc regeneration Timepoints: 1 and 3 months	Detiger et al. (2015) [127]
6 male goats, 6 months, 23–25 kg	Damage to the annulus fibrosus of the intervertebral discs T1 to L5, with a 1 × 1 cm^2^ gap	Treatment with a combination of BMSCs and PRP in a cell suspension with a gelatin sponge (1 × 1 cm^2^) Timepoints: 3, 6, and 12 weeks	Xu et al. (2019) [126]
3 adult male goats, 3 years	Injection of 1 U chondroitinase ABC in lumbar discs through a left retroperitoneal approach	Treatment with a hydrogel composed of dextran, chitosan, and teleostean for augmentation of the nucleus pulposus	Gullbrand et al. (2017) [125]
9 male goats, ±3 years	L1-2, L2-3, L3-4, and L4-5 lumbar discs were randomized to receive either subtotal nucleotomy, saline sham injection, or chondroitinase ABC (0.1 U, 1 U, or 5 U)	Evaluation of inflammatory cytokines (TNF-α, IL-1β, IL-6) and catabolic enzymes (MMPs-1 and 13, and ADAMTS-4) expression. Timepoint: 12 weeks	Zhang et al. (2020) [120]
10 mature male goats, 2–4 years	Injection of 1 U chondroitinase ABC in five lumbar discs (L1-L5) through a left retroperitoneal approach	Three groups treated with: - hydrogel alone, - combined hydrogel + MSCs, - untreated control. 2 weeks after treatment	Zhang et al. (2021) [119]

IVD—Intervertebral disc, PLCL—poly(L-lactide-co-caprolactone), BMSCs—bone marrow derived-mesenchymal stem cells, SVF—stromal vascular fraction, ASC—adipose-derived mesenchymal stem cells, PRP—platelet-rich plasma, TNF-α—tumor necrosis factor-alpha, IL-1β—interleukin 1-beta, IL-6—interleukin-6, MMPs—matrix metalloproteinases, ADAMTS-4—A Disintegrin and Metalloproteinase with Thrombospondin Motif-4.

## Data Availability

Not applicable.

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
