# Peer review of "Mesenchymal Stem Cell Studies in the Goat Model for Biomedical Research—A Review of the Scientific Literature"

_biology, 2022, doi:10.3390/biology11091276_

Round 1
Reviewer 1 Report
Dear authors,
The manuscript entitled "Mesenchymal stem cell studies in the goat model for biomedical research – A review of the scientific literature" explores the topic of the application of MSCs and the application of goats in research using a wide approach. The authors discussed several areas of medicine and health which may be possible to apply these models. The design of this narrative review is correct and well-structured. Tables and figures are well positioned in order to understand the past literature described on this topic. I have a few suggestions to increase even more the quality of this manuscript.
1- The authors can include a section with a search methodology, describing some keywords used to create this review and the range (years) that the articles reviewed were published. Also, which kinds of articles were included and excluded for this review.
2- In chapter 4.4., the authors explored well some applications of MSCs, however, I feel that currently, this approach is increasing and several articles describe the application of MSCs in biomaterials for bone regeneration and in dental implants in order to try to reproduce a more similar environment in in vitro research to the human body. Basic studies have applied these cells to test innovative surfaces and new biomaterials. The authors can explore this a little bit more in this chapter with more references. Some suggestions below:
- DOI: 10.1038/srep38814
- DOI: 10.1563/AAID-JOI-D-10-00206
- DOI: 10.11607/jomi.8069
3- The authors can create a chapter before the conclusion section describing the limitations of the application of goats in several areas of research. Size, regulations, costs, differences to the human body....
Author Response
The manuscript entitled "Mesenchymal stem cell studies in the goat model for biomedical research – A review of the scientific literature" explores the topic of the application of MSCs and the application of goats in research using a wide approach. The authors discussed several areas of medicine and health which may be possible to apply these models. The design of this narrative review is correct and well-structured. Tables and figures are well positioned in order to understand the past literature described on this topic. I have a few suggestions to increase even more the quality of this manuscript.
1- The authors can include a section with a search methodology, describing some keywords used to create this review and the range (years) that the articles reviewed were published. Also, which kinds of articles were included and excluded for this review.
ANSWER: We are very grateful for all the suggestions made that improved the quality of the manuscript, all of which were accepted and carried out. We have introduced the keywords used to create this review, the range of years that the articles reviewed were published and the type of articles that were included and excluded for this review – page 3 (lines 148-160).
2- In chapter 4.4., the authors explored well some applications of MSCs, however, I feel that currently, this approach is increasing and several articles describe the application of MSCs in biomaterials for bone regeneration and in dental implants in order to try to reproduce a more similar environment in in vitro research to the human body. Basic studies have applied these cells to test innovative surfaces and new biomaterials. The authors can explore this a little bit more in this chapter with more references. Some suggestions below:
- DOI: 10.1038/srep38814
- DOI: 10.1563/AAID-JOI-D-10-00206
- DOI: 10.11607/jomi.8069
ANSWER: We have introduced a brief reference in the text describing the application of MSCs in biomaterials for bone regeneration and in dental implants, in order to try to reproduce a more similar in vitro research environment to the human body. We have also introduced in the text the three articles suggested (DOI: 10.1038/srep38814, DOI: 10.1563/AAID-JOI-D-10-00206 and DOI: 10.11607/jomi.8069) – page 15 (lines 579-580 and 587-589). We added a new section in the manuscript with a brief overview of biomaterials for MSCs regenerative medicine and tissue engineering (pages 5 and 6).
3- The authors can create a chapter before the conclusion section describing the limitations of the application of goats in several areas of research. Size, regulations, costs, differences to the human body....
ANSWER: The limitations regarding the use of the goat model in research are mentioned, and reinforced, in section 2 of the manuscript "The goat model" – pages 3 and 4 (lines 188-207) and in table 1. Furthermore, in each of the areas in which the goat model is used in studies involving stem cells in orthopaedics, dermatology, ophthalmology, dentistry, pneumology, cardiology and urology, some more specific advantages or disadvantages, relating to different tissues or organ systems are mentioned depending on the similarities or differences in the biology of the goat species in relation to that of the human species.
Reviewer 2 Report
This manuscript summarize the characteristics of MSCs and the advantages and disadvantages of goat model, then review the latest research focused on studying the biological and therapeutic petential of MSCs in the goat model. This manuscript would be better if the following points had been noted.
1. As an important part of tissue engineering and regerative medicine, materials (natural or synthetic) applied with MSCs in goat models cannot be ignored. Actually, the materials can induce or inhibit the proliferation and differentiation of MSCs, which plays a critical role in wether MSCs can differentiate into the target cell types. Therefore, it is recommended that the authors discuss in detail the application of the materials and its impact on MSCs.
2. The authors described the disadvantages such as higher maintenance costs, harder to handle, requires specialized housing facilities infrastructure and trained personnel, and longer gestation time, as we all know, which limit preliminary studies with large samples. Compared with small animals (mouse, rabbits, frog, dog, etc.), goat models are not suitable for large-scale research phases of therapeutic technologies, but for preclinical preparations. The authors should explain this in the manuscript. In the other hands, the authors argue the advantages of goat models such as similarity of anatomy and dimensions to humans, similarity of physiology and clinical procedures to humans, but why not consider the pigs? Actually, as we all know, the pig gene sequence is highly homologous to that of humans.
3. The manuscript was no essential prospects for the application of MSCs as well as materials to goat models.
Author Response
This manuscript summarize the characteristics of MSCs and the advantages and disadvantages of goat model, then review the latest research focused on studying the biological and therapeutic petential of MSCs in the goat model. This manuscript would be better if the following points had been noted.
- As an important part of tissue engineering and regerative medicine, materials (natural or synthetic) applied with MSCs in goat models cannot be ignored. Actually, the materials can induce or inhibit the proliferation and differentiation of MSCs, which plays a critical role in wether MSCs can differentiate into the target cell types. Therefore, it is recommended that the authors discuss in detail the application of the materials and its impact on MSCs.
ANSWER: We are very grateful for all the suggestions made that improved the quality of the manuscript, all of which were accepted and tried to carried out. We introduce a new section in the manuscript with a brief overview of biomaterials for MSCs regenerative medicine and tissue engineering (pages 5 and 6). Having done so, we would still wish to reinforce that this review focuses on the use of stem cells in goat animal model; there are several different scientific publications addressing the application of different biomaterials and its specific interactions with cells (namely stem cells), which would constitute a review in itself. This was never the aim of our review.
- The authors described the disadvantages such as higher maintenance costs, harder to handle, requires specialized housing facilities infrastructure and trained personnel, and longer gestation time, as we all know, which limit preliminary studies with large samples. Compared with small animals (mouse, rabbits, frog, dog, etc.), goat models are not suitable for large-scale research phases of therapeutic technologies, but for preclinical preparations. The authors should explain this in the manuscript. In the other hands, the authors argue the advantages of goat models such as similarity of anatomy and dimensions to humans, similarity of physiology and clinical procedures to humans, but why not consider the pigs? Actually, as we all know, the pig gene sequence is highly homologous to that of humans.
ANSWER: We agree that the pig is a very attractive animal model for preclinical and translational studies in man, namely due to the pig gene sequence knowledge and for being highly homologous to humans. However, we carried out this bibliographic review specifically on the use of the goat model in tissue engineering and regenerative medicine studies involving MSCs in order to fit into the Special Issue of Biology journal under the theme “The application of the goat model in biomedical research”. However, now we have also introduced in the manuscript a brief reference to the MSCs surface marker expression patterns and the goat gene sequence to reinforce the utilization of the caprine model in tissue engineering and regenerative medicine studies involving MSCs – page 4 (lines 196-200).
- The manuscript was no essential prospects for the application of MSCs as well as materials to goat models.
ANSWER: In this review article, we have tried to compile the preclinical studies carried out in the last two decades in the goat model using MSCs, either alone or associated with materials. We found that they essentially aimed at repairing injuries of traumatic, disease or degenerative aetiology related to the musculoskeletal system, dermatology, ophthalmology, dentistry, pneumology, cardiology and urology fields. It appears that the goat is used in these studies, especially those related to the locomotor system, cardiology and urology, essentially due to the size, active behavior and anatomical characteristics of the goat that are similar to humans. The goat knee anatomy and the thickness of the cartilage that covers this joint is closer to that of humans than that of other large animal models commonly used in orthopedic research. Also, the goat model appears to be extremely useful for the study of cardiac and urologic diseases, such as myocardial infarction and stress urinary incontinence due to the similarity in the anatomy of the goat heart and urethra, respectively. It is not so much the typology of MSCs or materials that condition the use of this animal model in tissue engineering and regenerative medicine studies, but rather its anatomical, physiological and behavioral characteristics. These aspects are mentioned throughout the manuscript and in its conclusion.
Reviewer 3 Report
The manuscript presented by Inês E. Dias et al., gives a thorough overview on the latest research focused on studying the biological and therapeutic potential of MSCs in 30 the goat mode.
Overall, the manuscript is well organized and very informative. My only comment is that it might benefit from some language editing and a thorough revision is required for typos and un-scientific terms.
Author Response
The manuscript presented by Inês E. Dias et al., gives a thorough overview on the latest research focused on studying the biological and therapeutic potential of MSCs in the goat mode.
Overall, the manuscript is well organized and very informative. My only comment is that it might benefit from some language editing and a thorough revision is required for typos and un-scientific terms.
ANSWER: We are very grateful for the general evaluation of the manuscript, and we carry out an English language review performed by a native English-speaking colleague.
Round 2
Reviewer 1 Report
The authors improved several details of this manuscript. The review is very complete and it has all the necessary guidelines for a narrative review, having merit for publication.
Best regards.